# Cost-Effectiveness of Titanium Elastic Nail (TEN) in the Treatment of Forearm Fractures in Children

**DOI:** 10.3390/medicina56020079

**Published:** 2020-02-15

**Authors:** Ovidiu Adam, Vlad Laurentiu David, Florin George Horhat, Eugen Sorin Boia

**Affiliations:** 1Department of Pediatric Surgery and Orthopedics, “Victor Babes” University of Medicine and Pharmacy, Eftimie Murgu Sq. No 2, 300041 Timisoara, Romania; adamovidiu29@yahoo.com (O.A.); boiaeugen@yahoo.com (E.S.B.); 2Department of Microbiology, “Victor Babes” University of Medicine and Pharmacy, Eftimie Murgu Sq No 2, 300041 Timisoara, Romania

**Keywords:** forearm fractures, children, fracture treatment, titanium elastic nail, Pediatric orthopedics, cost analysis

## Abstract

*Background and objectives*: There are various methods in the management of forearm fractures in children. Elastic stable intramedullary nailing using Titanium Elastic Nail (TEN) is nowadays employed in diaphysis fractures of children, with clear benefits over other treatment options. However, in the case of TEN versus other treatment methods of forearm fractures in children, cost is an important issue. This report will focus on the cost assessment of using TEN versus other therapeutic means in the treatment of forearm fractures in children. *Materials and Methods*: We performed a retrospective longitudinal study of 173 consecutive patients with forearm fractures treated in a single institution during 2017. We calculated the cost for each patient by summing up direct costs plus indirect costs, calculated at an aggregate level. Hospital income data were extracted from the Diagnosis Related Groups database. *Results*: A total of 173 patients with forearm fractures were treated, 44 using TEN, 86 using K-wire, and 46 using closed reduction and cast. There were 66 radius fractures, 1 ulna fracture, and 106 that were both radius and ulna fractures. Mean treatment cost were $632.76 for TEN, $499.50 for K-wire, and $451.30 for closed reduction and cast. Costs for TEN were higher than for K-wire insertion (*p* = 0.00) and higher than closed reduction and cast ($182.42; *p* = 0.00). Reimbursement per patient was higher with TEN versus K-wire patients; $497.88 vs. $364.64 /patient (*p* = 0.00), and higher than for patients treated with closed reduction and cast (*p* = 0.00). *Conclusions*: The treatment of upper extremity fractures using TEN was more expensive than the other methods. In Romania, because the reimbursement for TEN is higher as well, there are no differences in the financial burden when treating forearm fractures with TEN versus K-wire. Non-surgical treatment has the lowest cost but also the lowest reimbursement.

## 1. Introduction

Forearm fractures are the most common fractures in childhood [1]. There are various methods of management of forearm fractures in children: Closed reduction and plaster cast, closed/open reduction, and intramedullary K wire stabilization. Elastic intramedullary nails were introduced for the first time in the late 1970s, and since then, the technique has changed very little [2]. Initially, it was used for children for whom cast treatment was not a viable solution but had soon been adopted for all diaphysis fractures. Elastic stable intramedullary nailing (ESIN) using Titanium Elastic Nail (TEN) or Stainless-Steel Nail (STEN) is nowadays employed for diaphysis fractures and metaphysis fractures for children, unstable fractures of the forearm, and some of the unstable fractures of the humerus and polytrauma, and for multiple injuries [3,4,5,6,7,8]. Due to its success, it has been introduced for several additional types of fractures: metaphysis, fragmented, pathological bone fracture, and fractures of small bones (such as the collarbone and metacarpal bones) [5,6,7,8,9,10,11]. The main advantages are: it is a minimally invasive approach resulting in closed reduction and preservation of the fracture hematoma. Postoperative cast can be avoided so that early mobilizations is possible [4,10].

For any medical system, cost is a major issue in the decision to adopt one treatment option over another [12]. In the case of TEN versus other options for the treatment of long bone fractures in children, cost is also an important issue. Only a few publications have focused on the financial aspects of using TEN in children, and most of them focused on fractures of the lower limb long bones [13].

This report will focus on the cost assessment of using TEN versus other therapeutic means in the treatment of forearm fractures in children.

## 2. Materials and Methods

We performed a retrospective longitudinal study of 173 consecutive patients with forearm fractures treated in a single institution during 2017. The study was conducted at the “Louis Turcanu” Children’s Hospital, Timisoara, Romania. Prior approval was granted by the Ethical Committee of the “Louis Turcanu” Children’s Hospital, Timisoara, Romania (No. 120/12, June 2019). We scanned the hospital database and recorded data regarding: age, gender, diagnosis, comorbidities, treatment method, complications, and hospital stay. We calculated the cost for each patient according to hospital controlling principles, using controlling methodology. The cost per patient was calculated by summing up direct costs (medication, surgical materials) and indirect costs calculated at an aggregate level (overheads, diagnostic costs, hospital management, service and maintenance of the equipment, and salaries of the medical personnel).

Hospital income data were extracted from the Diagnosis Related Groups database comprising all patients treated during 2017. The income for each patient, received by the hospital from the National Health Insurance Fund (NHIF) was calculated by multiplying the Case Mix Index (CMI) with the standard tariff per patient. The cost for K-wires and TEN are covered on separate databases by the National Health Program of Trauma (NHPT), so when we calculated the reimbursement for each patients, we added the direct costs of TEN ($71/TEN) and those of K-wires ($1.2) separately to the financial support received from the NHIF.

The cost calculation was performed in the local currency (RON) and converted to United States Dollars at the exchange rate of the National Bank of Romania ($1 = 4.2268 RON as at 23.04.2019)

Cost comparison was performed using the unpaired *t*-test with a significance threshold set at *p* = 0.05 for 95% CI. We used Pearson’s product-moment correlation to calculate if there was a correlation between the different parameters.

## 3. Results

From 1 January to 31 December, 2017 a total of 173 patients (45 girls and 128 boys) with forearm fracture(s) were admitted and treated in our hospital. The age of the patients ranged from 3.3 to 19.5 years (mean 12.1 years). There were 66 radius fractures, 1 ulna fracture, and 106 that were both radius and ulna fractures (Table 1). Closed reduction and cast was used in 46 patients, closed or open reduction and K wire(s) was used in 82 patients, and closed or open reduction and TEN was used in 44 patients. Postoperative cast was used in all of the patients treated with K-wires, in 5 patients treated with 1 TEN, and none of the patients treated with 2 TEN.

The mean length of stay was 3.43 days (1–8 days); 3.57 for TEN patients, 3.55 days (1–7 days) for K-wire patients, and 3.09 days (1–6 days) for closed reduction and cast patients. Only 3 patients with polytrauma were admitted into the Intensive Care Unit.

The mean cost for treating forearm fractures was $520.09 ($337.43–$455.53)/patient (Table 2). The mean cost for TEN was higher than for K-wire insertion (mean difference, $131.80; *p* = 0.00) and higher than closed reduction and cast (mean difference, $182.42; *p* = 0.00). The cost for treatment with K- wire was also higher than closed reduction and cast (mean difference, $50.70; *p* = 0.03).

Reimbursement per patient was higher in TEN versus K-wire patients; $497.88 vs. $364.64 per patient (mean difference, $131.16, *p* = 0.00), and higher than for patients treated with closed reduction and cast (mean difference, $343.92; *p* = 0.00). The balance expenditure per income per patient was negative for all treatment methods; TEN, K-wire, and closed reduction and cast. Mean loss per patient was similar (*p* > 0.05) for TEN patients; $133.26 ($16.29–$700.94) and K-wire patients; $132.62 (−$0.87–$722.58) per patient, and was higher (mean difference, $178.61, *p* = 0.00) for closed reduction and cast patients; $311.91 ($146.01–$591.93).

TEN wase removed in 36 patients and K-wire in 53 patients. Mean cost for TEN removal was higher than for K-wire removal (*p* = 0.002) (Table 2). Reimbursement for TEN and K-wire removal was similar (*p* > 0.05), and the balance expenditure/income per patient was positive in both TEN and K-wire removal: $621.15 (−$22.09–$558.99) for TEN and $619.61 (−$504.85–$1496.88) for K-wire removal.

Complication occurred in 17 patients; 1 treated with TEN, 13 with K-wire, and 3 treated with closed reduction and cast, but these had no impact on cost (*p* > 0.05). Only 3 patients had sustained polytrauma and were admitted to the Intensive Care Unit. This had a direct impact on the costs (mean difference, $247.89, *p* = 0.01).

## 4. Discussion

In Romania, inpatient care is financed by the National Health Insurance Fund (NHIF) via the Diagnosis Related Groups system. Despite the fact that the budget for health care financing shows a positive trend, the resources available for financing the healthcare system are limited, therefore cost analysis is gaining more and more importance both at national and at international levels [13,14].

Our results show that intramedullary nailing of upper limb fractures using TEN is more expensive than other fracture stabilization methods. These results are consistent with those of previous studies [15,16,17,18]. However, additional factors besides the direct hospital costs have to be considered when choosing the optimal treatment method. It is a known fact that TEN stabilization of upper limb fracture has a better medical and functional outcome [19,20]. In our series, most of the patients treated with TENs did not require a postoperative cast at all; while all patients being treated by the other methods (K-wire or closed reduction and cast) had postoperative cast immobilization for several weeks, meaning prolonged impairment for those children. Similar to other fractures, such as lower limb fractures, sometimes additional healthcare assistance, such as rehabilitation, has to be considered, meaning increased financial burden for the health system [21,22]. Beside the additional costs of care and that of healthcare materials, there are other costs to be considered in upper extremity fractures in children. When analyzing the differences in healthcare costs, Shore et al. [16] considered the financial loss caused by the decrease in work productivity of the patients’ parents and also other social expenses or direct costs for the patients (for example, the costs of postoperative complications) or for their parents, such as transport costs [15]. Even though not directly covered by the medical system, these kinds of costs represent a burden to the families and society.

Cost reduction can be achieved by different approaches. Hospital stay and intensive care unit (ICU) stay are one of the main cost generators in our hospital [12]. Cost reduction may be achieved by reducing the admission period to the minimum necessary, even to one-day surgery. This is the standard of care in many hospitals around the world [3,5,19,20]. Furthermore, modifications to the intramedullary nailing technique were introduced by several authors: Some of the authors routinely use a single nail stabilizing both the forearm bones, with comparable results to using the classic two nail insertion technique, or recommend careful consideration when choosing the surgical technique and materials for fracture stabilization [18,23]. On the other hand, when trying to reduce the cost, the entire picture, and not only the cost generated by the inpatient treatment, has to be considered. Unfortunately, this is a limitation for our study, because the outpatient care and outpatient-based hospital expenses and incomes could not be assessed properly. Previous studies have demonstrated that this outpatient medical assistance is a significant generator of cost as well, and most probable due to increased necessity for visiting physicians, dressing, and cast removal. Patients treated by other methods than TEN have an increased need for outpatient medical assistance (13).

The other major factor influencing the balance between hospital income and expenses is the amount of reimbursement per patient. In Romania, the reimbursement for treating fractures is made from two different systems: Ministry of Health covers the cost of the implants through the NHPT, while the NHIF reimburse the rest of the cost through the DRG system. Through the DRG system, the reimbursement from NHIF is similar, regardless of using TEN or K-wire to stabilize the forearm fracture. When using closed reduction and cast, the amount of reimbursement is lower than in TEN and K-wire. In our study, there was a negative balance regardless of the method of treatment. Even though the costs are higher in TEN patients, the deficit was similar to those treated by K-wire implants, the differences being covered by the NHPT. This means that the only differences in cost when treating patients with forearm fractures with TEN and K-wire is the cost of the implants. Paradoxically, reimbursement from Romanian NHIF was higher, generating a positive balance when the children were admitted for removal of the TEN or K-wires. This is an anomaly of the system, but it fortunately has the fundamental role of balancing the expenditure: Income equation in the end. Completing the treatment of patients by removing the implants through the inpatient system is bringing balance to the cost per patient and avoiding the negative balance per patient. Unfortunately, in only a fraction of the patients (≈50%), the implant was removed through the inpatient system, meaning that the deficit from the insertion of the implants were to be covered by the second admission only in half of the patients. On the other hand, K-wires can be removed without anesthesia in the outpatient clinic, particularly when leaving K-wire ends protruding over the skin. This situation implies supplementary costs because of the necessity of frequent wound dressing or the possibility of local infections, meaning more frequent visits to the outpatient unit. Moreover, these costs are not covered by the inpatient reimbursement system. In patients treated by closed reduction and cast, the financial loss per patient is permanent because removing the cast does not require hospital admission. However, reimbursement is not similar across countries, so our results are with regards to the peculiarity of the Romanian NHIF reimbursement system. The basic fact remains that the treatment of forearm fractures with TEN is more expensive than the other methods.

## 5. Conclusions

The treatment of forearm fractures using TEN is more expensive than using K-wire stabilization or non-surgical treatment. Mean treatment costs are $632.76 for TEN, $499.50 for K-wire, and $451.30 for closed reduction and cast. In our series, there is a negative balance reimbursement versus expenditures for each treatment method, with the mean loss being similar (≈$130) for TEN and K-wire and higher for patients treated by closed reduction and cast (≈$300). There is a positive balance (≈$600) per patient with the removal of both TEN and K-wire, covering most of the financial deficit from the insertion of the implants. The financial deficit could not be recovered for the patients treated by closed reduction and cast, and we recommend not doing this through the inpatient system. Based on these facts, we can affirm that, in Romania, there are differences in cost in the treatment of forearm fractures with TEN versus K-wire, but there are no differences in the financial burden for the hospital of one treatment method over the other.

## Figures and Tables

**Table 1 medicina-56-00079-t001:** Diagnostic versus treatment.

	No.	CR + Cast	1 K-wire	2 K-wire	1 TEN	2 TEN	PO Cast
Radius	66	30	29	7	7	0	29
Ulna	1	0	0	0	1	0	1
Radius + Ulna	106	16	52	1	2	34	53
Total	173	46	81	1	10	34	83
			82		44		

No: number of patients; CR: closed reduction; TEN: titanium elastic nail; PO cast: postoperative cast.

**Table 2 medicina-56-00079-t002:** Hospital expenditures and reimbursement.

Procedure	Expenditures ($)	Reimbursement ($)	TEN/K-Wire Removal ($)	Reimbursement Removal ($)
TEN	471.16–1073.00 Mean 632.76	443.03–514.01 Mean 497.88	385.30–672.54 Mean 462,84	512.76–1867.47 Mean 1066.61
K-wire	372.36–1095.82 Mean 499.50	162.16–455.53 Mean 364.64	365.05–636.01 Mean 424.71	107.31–1867.47 Mean 1044.32
CR + Cast	337.43–699.25 Mean 451.30	107.31–372.06 Mean 150.03	-	-
Overall	337.43–455.53 Mean 520.09	107.31–455.53 Mean 309.51	365.05–672.54 Mean 439.52	107.31–1867.47 Mean 1043.91

CR: closed reduction; TEN: titanium elastic nail.

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
