# Peer review of "Cost-Effectiveness of Titanium Elastic Nail (TEN) in the Treatment of Forearm Fractures in Children"

_medicina, 2020, doi:10.3390/medicina56020079_

Round 1

Reviewer 1 Report

This study presents the costs of  the treatment of forearm fractures in children.

Thank you for re-submitting this work. It has significantly improved from the previous version and is now easy to follow.

Compared to the previous version, I am happy to find the description of the other treatment methods as well.

Author Response

Thank you!

Reviewer 2 Report

The study design must be improved. You compared groups which are not comparable. Why they have different night permanance in hospital. You do not compare clinical outcomes. Age is not described. You do not report any complication of the devices. You just compare cost material and rembourse.

Author Response

Dear Sir/ Madame

Thank you for your opinion

The declared purpose of this study is to compare costs of using TEN versus other therapeutic means in the treatment of forearm fractures in children. We did not compared the clinical outcomes nor the choice for the treatment methods. There are several studies demonstrating that TEN is the best choice. We wanted to find out if using TEN means extra financial burden to the Hospital/ Health System

Age is described at Results, line 81-82. It was not a factor influencing the cost for the treatment and there were no difference among groups. If you consider necessary, we may add this in the text.

Since it is not a report of the clinical outcomes and since we had no major complications, we reported the complications from the financial point of view. Results, line 107-108

We considered that the groups are comparable from the clinical and financial point of view. The lot is homogenous, the patients had the same type of pathology for which different types of treatment methods were used.

This manuscript is a resubmission of an earlier submission. The following is a list of the peer review reports and author responses from that submission.

Round 1

Reviewer 1 Report

I believe the report would benefit from briefly describing the patients and the type of fractures to bring the reader infromation of the type of patients that were treated.

The description of the other treatment method was inadequate.

Reviewer 2 Report

I would like to congratulate the authors on their efforts. This study performed cost and income measurements of using Titanium Elastic Nail (TEN) vs. other surgical stabilization techniques for treatment of upper extremity fractures

The authors report that, the treatment of upper extremity fractures using TEN was more expensive than the treatment using other methods. The mean length of stay, the mean length of stay in Intensive Care Unit (ICU) and the mean length of the surgical intervention were found to be longer for TEN compared with other treatment procedures.

I agree with the authors that cost analysis is gaining more and more importance. Therefore, this is an important study that can be useful as a basis for evaluating the treatment of upper extremity fractures in children. However, based on my comments below, I can not recommend publication of the manuscript.

Major Recommendations:

My main comment is that the manuscript does not evaluate weather or not TEN is a cost effective treatment. Maybe I totally misunderstood the types of fractures in the two groups. The manuscript doesn't provide a demographic table comparing the two groups (TEN vs Other methods). What types of fractures was included? Comparing the cost of a K-wire treated distal humerus fracture or a distal radius fracture with a TEN treateddiaphyseal radius/ulna fracture can be of interest but in that case I believe the result needs to be presented in another way.

Does the study have an institutional review board approval?

Specific comments:

Abstract:

The introduction states that "The principle underlying the procedure is that two nails are introduced into the intramedullary canal....". This is true but for forearm fractures the principle is a single nail inserted in ulna and radius.

The methods state "....cost calculations included direct as well as indirect costs." I agree that the study have calculated direct costs but I do not understand how indirect costs (Parental productivity loss during the period of their child’s injury) was estimated. 

The result does not include any results supporting the conclusion.

The introduction:

Line 38 "gypsum" could be changed to "cast" The paragraph about TEN should focus more on the treatment of different upper extremity fractures. I am missing an paragraph regarding other types of surgical treatment for different upper extremity fractures. Line 53 Please provide a reference supporting the use of TEN in adults. What is the hypothesis of the study?

Methods:

How was the patients identified? For me direct costs refers to all costs due to the use of health care for the fracture. Indirect costs refers to work loss, worker replacement, and reduced productivity from illness and disease (Parental productivity loss during the period of their child’s injury). I can not follow how indirect costs are collected. Please provide a statistical paragraph. Does the study have an institutional review board approval?

Results:

I am missing time from admission to surgery when calculating lenght of stay. In my experience most children can leave the hospital the day after an uncomplicated surgery with TEN for a forearm fracture. In my hospital removal of TENS is done as a day surgery but I can understand that the patients for practical reasons have to be admitted. I was therefore very surprised that the result stated: "The mean length of stay was 3.95 days (range: 2-12 days), 4.29 for TEN insertion and 2.89 for TEN removal." Any post operative complications? Did you include cost for outpatient controls after surgery? How many patients in each group were treated in the ICU? This cost should be presented for each group since time in ICU probably is more related to other injuries than to the treatment. I believe the tables could be more detailed and improved. Please provide data for different types of upper extremity fractures.

Discussion:

This section is very short, and I would like it to be expanded. I am also missing a section describing the limitations and strengths of the present study.

Conclusion:

- I believe this section should be rewritten since this I believe the study does not evaluate weather or not TEN is a cost effective treatment.